# From Deep Learning to Deep Deducing: Automatically Tracking Down Nash Equilibrium through Autonomous Neural Agent, a Possible Missing Step toward General A.I.

## Abstract

Contrary to most reinforcement learning studies, which emphasize on training a deep neural network to approximate its output layer to certain strategies, this paper proposes a reversed method for reinforcement learning. We call this "Deep Deducing". In short, after adequately training a deep neural network according to a strategy-environment-to-payoff table, then we initialize randomized strategy input and propagate the error between the actual output and the desired output back to the initially-randomized strategy input in the "input layer" of the trained deep neural network gradually to perform a task similar to "human deduction". And we view the final strategy input in the "input layer" as the fittest strategy for a neural network when confronting the observed environment input from the world outside.

## 1 Introduction

Contrary to most reinforcement learning studies (as in "Playing Atari with Deep Reinforcement Learning (Mnih et al., 2013)."), which emphasize on training a deep neural network to approximate its output layer to certain strategies and view the output layer of a deep neural network as its strategy in a certain natural environment, this paper provides a revolutionary way for reinforcement learning and a possible road toward general A.I..

Also, contrary to most Game Theory studies, which emphasize on the role of human experts to device smart mathematics algorithm (as in "When Machine Learning Meets AI and Game Theory (Agrawal & Jaiswal, 2012)."), this paper emphasizes on the role of machine and tends to show a bridge between A.I. territory and Game Theory.

The method is simple. In short, after training a deep neural network according to a strategy-environment-to-payoff (input-to-output) table that records possible strategies and the environments along with the consequential payoffs, then we randomize the strategy input in the "input layer" and propagate the error between the actual output and the desired output back to the strategy input in the "input layer" of the same trained deep neural network recurrently to perform a task similar to "human deduction". And we view the final strategy input in the "input layer" as the fittest strategy for a neural network confronting the observed environment input from the world outside. Since the information is sent back to the input layer in a reversed way like deduction in logic, we can call this "Deep Deducing".

By using the same trick, we can force a neural network to automatically track down the Nash Equilibrium in a Simultaneous Discrete Game (which will be explained later). Since Nash Equilibrium is the optimal combination of strategies from which no player will be willing to deviate, and, for any player who deviates from NE, the player will be punished, the ability to find the Nash Equilibrium in advance in a randomized complex system indicates certain intelligence in an autonomous agent.

## 2 ALGORITHM: DEEP DEDUCING

To grasp the intuition behind Deep Deducing, we will first take a mosquito for example as shown in Figure 1.

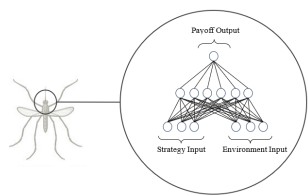

Figure 1: example of a simplified brain of a mosquito

Imagine the brain of a mosquito consists of a canonical deep feedforward neural network with only one input layer, one hidden layer, and one output layer. The input layer consists of strategy input and environment input. And the output layer consists of payoff out. A list of neurons in the strategy input records the strategies (denoted $\mathbf{i}^s$) that the mosquito has adopted in response to certain environment information (denoted $\mathbf{i}^n$), which environment information is also simultaneously recorded by the other list of the neurons in the environment input as well. Meanwhile the neurons in the payoff output also instantaneously record the payoff (denoted $\mathbf{o}$) to the mosquito when it adopts a certain strategy $\mathbf{i}^s$ under a certain environment $\mathbf{i}^n$. The concatenation of $\mathbf{i}^s$ and $\mathbf{i}^n$ is denoted as $\mathbf{i}$.

The brain of the mosquito is trained to memorize the set of concatenations of strategy and environment (denoted $\mathbb{I}$) as well as the set of their payoffs (denoted $\mathbb{O}$). That is the set of the weights of the synapse (denoted $\mathbb{W}$) are adjusted through (Stochastic Gradient Descent and Back Propagation):

$$\mathbb{W}_{t+1} \leftarrow \mathbb{W}_t - \alpha \frac{\partial}{\partial \mathbb{W}_t} E(\mathbf{o} - \mathbf{o}') \tag{1}$$

where $t$ is the current epoch, $\alpha$ is the learning rate, $E$ is the error function (the loss or cost function, here we use squared error), and $\mathbf{o}'$ is the estimated payoff generated by current $\mathbf{i}$ and $\mathbf{W}_t$. This is Deep Learning.

For illustration, the consequence of seeing human's hand and choosing not to fly away is recorded. Of course the consequence is death, in which case the payoff to the mosquito is [0]. A cloud (of IoT) records the consequence (payoff) of every combination and trains the brain (the neural network) of the mosquito according to Figure 2 as below.

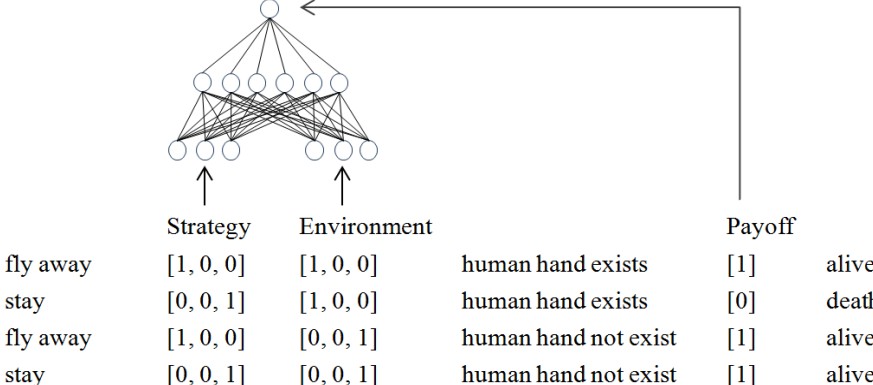

|  | Strategy | Environment |  | Payoff |  |
|---|---|---|---|---|---|
| fly away | $[1, 0, 0]$ | $[1, 0, 0]$ | human hand exists | $[1]$ | alive |
| stay | $[0, 0, 1]$ | $[1, 0, 0]$ | human hand exists | $[0]$ | death |
| fly away | $[1, 0, 0]$ | $[0, 0, 1]$ | human hand not exist | $[1]$ | alive |
| stay | $[0, 0, 1]$ | $[0, 0, 1]$ | human hand not exist | $[1]$ | alive |

Figure 2: example of the learning process in a mosquito's brain

When the brain (the neural network) of the mosquito is adequately trained (by using Stochastic Gradient Descent) according to the table in Figure 2, upon seeing a human's hand at presence, the brain (the neural network) of the mosquito must figure out and make deduction about what strategy it shall adopt in order to stay alive. ***In another word, how could the neural network figure out a strategy that, when put into the neural network along with environment input [1, 0, 0], can generate the payoff [1]?***

The answer that Deep Deducing gives is rather simple – Just randomize the strategy input in the input layer of the trained neural network and let the neurons that represent the strategy input to adjust itself gradually to render the neural network to reach output [1] by propagating the error between the generated output and the desired output [1] back to the neurons in the strategy input itself for fixed epochs or as long as the generated output (of the adjusted strategy input obtained in the last epoch) does not adhere to [1] in a reasonable deviation.

That is, for a given $\mathbf{i}^n$ where $\mathbf{0} \leq \mathbf{i}^n \leq \mathbf{1}$, a set of trained $\mathbb{W}_{trained}$ and a pre-fixed desired payoff $\mathbf{o}^d$, $\mathbf{0} \leq \mathbf{o}^d \leq \mathbf{1}$, we first initialize a randomized strategy input $\mathbf{i}_0^s$ at time zero where $\mathbf{0} \leq sigmoid(\mathbf{i}_0^s) \leq \mathbf{1}$. Then, by using Back Propagation, we let:

$$\mathbf{i}_{t+1}^{strategy} \leftarrow \mathbf{i}_t^{strategy} - \alpha \frac{\partial}{\partial \mathbf{i}_t^{strategy}} E(\mathbf{o}^d - \mathbf{o}_t') \tag{2}$$

where $t$ is the current epoch, $\alpha$ is the deducing rate (like learning rate), $E$ is the error function, and $\mathbf{o}_t'$ is the estimated payoff generated by current $sigmoid(\mathbf{i}_t^s)$, $\mathbf{i}^n$ and $\mathbb{W}_{trained}$. This is "Deep Deducing". Please notice that, $\mathbf{i}^n$, $\mathbb{W}_{trained}$ and $\mathbf{o}^d$ are fixed, and the only thing that is changing is $\mathbf{i}_t^s$ and $\mathbf{o}_t'$. And there is no Stochastic Gradient Descent since the desired payoff $\mathbf{o}^d$ is pre-fixed. ***By exploiting the existing trained neural network and the trick of Back Propagation, Deep Deducing forms a single, compounded and self-completed optimization problem for $i_t^s$ to reach the global minimum error, and therefore any kind of optimization skill, such as gradient-based particle swarm optimization, can be applied here as well***. For simplicity, we will only use the most plain and modest gradient descent with only one starting point for $\mathbf{i}_t^s$ to force the machine to track down Nash Equilibrium.

The whole process is illustrated in Figure 3.

1. Generate output from current strategy input obtained from step 0 or step 2 in the last epoch

2. Propagate the error between [1] and the generated output back to the strategy input and adjust the current strategy input by one epoch. And then repeat from step 1.

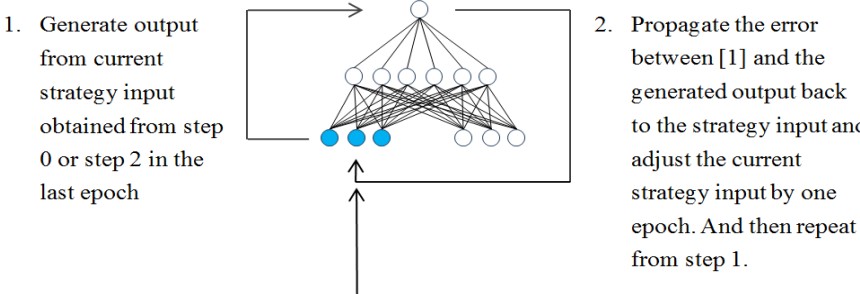

0. Initially randomize neurons that represent strategy input

*** The desired output, the neurons that represent the environment input and the weight of the synapse are all intact and left unchanged during all the epochs ***

| times | strategy | environment | actual payoff | desired payoff |
|---|---|---|---|---|
| 0 | $[0.4, 0.5, 0.6]$ | $[1, 0, 0]$ | $[0.3]$ | $[1]$ |
| 100 | $[0.5, 0.5, 0.5]$ | $[1, 0, 0]$ | $[0.3]$ | $[1]$ |
| 200 | $[0.6, 0.5, 0.3]$ | $[1, 0, 0]$ | $[0.4]$ | $[1]$ |
| 300 | $[0.8, 0.6, 0.2]$ | $[1, 0, 0]$ | $[0.5]$ | $[1]$ |
| … | … | … | … | … |
| N | $[0.9, 0.6, 0.1]$ | $[1, 0, 0]$ | $[0.99]$ | $[1]$ |

Figure 3: example of the deduction process in a mosquito's brain

Finally, after fixed epochs, according to the preliminary experimental result shown in Figure 3, the mosquito will obtain a strategy input that approximate [1, 0, 0] when argmaxed and one-hotted, which means the mosquito better just flies away. This final strategy information will later be sent to other parts of the mosquito to initiate the procedure of flying such as twitching wing tensors. We speculate that this simple three-layered deep neural network constitutes the very core of the mosquito's decision-making mechanism.

Therefore, the Deep Deducing algorithm can be dissected into two phases. The first is the training or learning phase, and the second is the deduction phase. In the training or learning phase, the mosquito learns about the consequence in response to the combination of strategy and environment. In the deduction phase, the mosquito must figure out a strategy that can make it stay alive when human is at presence. The pseudo code of Deep Deducing for the learning phase is shown in Algorithm 1.

---

**Algorithm 1** : training phase for the mosquito example

1: $\mathbb{I} \leftarrow set\, of\, strategies\, and\, environments$
2: $\mathbb{O} \leftarrow set\, of\, corresponding\, payoffs$
3: $\mathbb{W}_0 \leftarrow initialize\, set\, of\, randomized\, weights\, at\, time\, zero$
4: $\alpha \leftarrow learning\, rate$
5: $N_{epochs} \leftarrow number\, of\, epochs$
6: $N_{samples} \leftarrow number\, of\, training\, samples$
7: **for** $t$ **in** $[N_{epochs}]$ **do**
8: $\quad index \leftarrow randomint(N_{samples})$
9: $\quad \mathbf{i} \leftarrow \mathbb{I}[index]$
10: $\quad \mathbf{o} \leftarrow \mathbb{O}[index]$
11: $\quad \mathbf{o}' \leftarrow feedforward(\mathbf{i}, \mathbb{W}_t)$
12: $\quad \mathbb{W}_{t+1} \leftarrow \mathbb{W}_t - \alpha \frac{\partial}{\partial \mathbb{W}_t} E(\mathbf{o} - \mathbf{o}')$
13: **end for**
14: $\mathbb{W}_{trained} \leftarrow \mathbb{W}_{N_{epochs}}$

---

As for the deduction phase, the pseudo code can be shown in Algorithm 2.

---

**Algorithm 2** : deduction phase for the mosquito example

1: $\mathbf{i}_0^{strategy} \leftarrow initialize\, random\, strategy\, input\, at\, time\, zero$
2: $\mathbf{i}^{nature} \leftarrow environment\, input$
3: $\mathbf{o}^d \leftarrow initialize\, desired\, payoff\, output$
4: $\alpha \leftarrow deducing\, rate$
5: $N_{epochs} \leftarrow number\, of\, epochs$
6: **for** $t$ **in** $[N_{epochs}]$ **do**
7: $\quad \mathbf{o}'_t \leftarrow feedforward(concatenate(sigmoid(\mathbf{i}_t^{strategy}), \mathbf{i}^{nature}), \mathbb{W}_{trained})$
8: $\quad \mathbf{i}_{t+1}^{strategy} \leftarrow \mathbf{i}_t^{strategy} - \alpha \frac{\partial}{\partial \mathbf{i}_t^{strategy}} E(\mathbf{o}^d - \mathbf{o}'_t)$
9: **end for**
10: $optimal\, strategy \leftarrow onehot(argmax(sigmoid(\mathbf{i}_{N_{epochs}}^{strategy})))$

---

For the sake of simplicity, we will summarize the deduction phase in just one single sentence - ***Tuning the neurons in the strategy input to generate the desired output***.

## 3 EXPERIMENT: TRACKING DOWN THE NASH EQUILIBRIUM IN A SIMULTANEOUS DISCRETE GAME

The first experiment here we will examine is simply the extension of the notion of the mosquito example stated above.

If we take the mosquito and the hand of human as the two players A and B in a Simultaneous Discrete Game in Game Theory. We can train a neural network according to the strategies-to-payoff

table for player A and B, then we can tune the neurons in the strategy input for player A to generate the desired output for player A, and vice versa. After we interchangeably and iteratively tune these two sets of neurons each represents the strategy input for player A and B, these two sets of neurons will converge to the Nash Equilibrium.

For example, in Simultaneous Discrete Game, we often see the strategies-to-payoff table for player A and B as shown in Table 1.

|  | | B Strategies | |
| --- | --- | --- | --- |
|  | | $[1, 0, 0]$ | $[0, 0, 1]$ |
| A Strategies | $[1, 0, 0]$ | $[5, 5]$ | $[10, 0]$ |
|  | $[0, 0, 1]$ | $[0, 10]$ | $[7, 7]$ |

Table 1: table of prisoner's dilemma

In the same manner, we first train a neural network according to the strategies-to-payoff table for player A and B as illustrated in Figure 4 below.

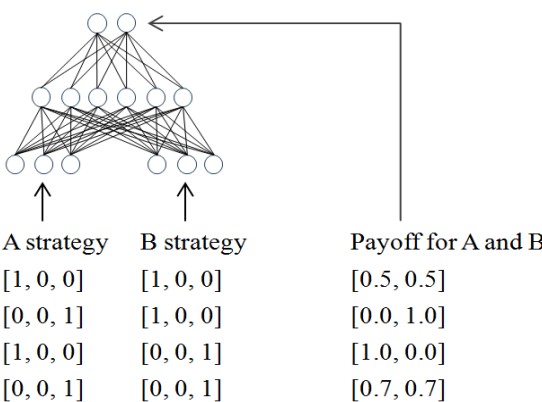

| A strategy | B strategy | Payoff for A and B |
| --- | --- | --- |
| $[1, 0, 0]$ | $[1, 0, 0]$ | $[0.5, 0.5]$ |
| $[0, 0, 1]$ | $[1, 0, 0]$ | $[0.0, 1.0]$ |
| $[1, 0, 0]$ | $[0, 0, 1]$ | $[1.0, 0.0]$ |
| $[0, 0, 1]$ | $[0, 0, 1]$ | $[0.7, 0.7]$ |

Figure 4: example of the learning process for prisoners

Second, since it is not a zero-sum game, we will tune (by one epoch) the neurons which represent the strategy input for player A to generate the desired payoff output for player A, namely [1, *] (where "*" stands for whatever the payoff for player B already is), and then we will tune (by one epoch) the neurons which represent the strategy input for player B to generate the desired output for player B, namely [*, 1] (where "*" stands for whatever the payoff for player A already is). We will keep this procedure for fair epochs as shown in Figure 5.

After fair epochs, as the preliminary experimental result in Figure 5, the neurons which represent the strategy input for player A will converge to [1, 0, 0] when argmaxed and one-hotted. And the neurons which represent the strategy of player B will also converge to [1, 0, 0] when argmaxed and one-hotted. This means that player A and player B will both adopt strategy [1, 0, 0], which is exactly the Nash Equilibrium in the strategies-to-payoff table in Table 1. The more formal experiment result is shown in Figure 9.

In Game Theory, this strategies-to-payoff table is Prisoner's Dilemma, and strategy input [1, 0, 0] is the "betrayal" strategy. Deep Deducing automatically tracks down the Nash Equilibrium in Prisoner's Dilemma which is that both players will betray each other (as in "Primer in Game Theory (Gibbons, 1994)."). Since Nash Equilibrium is the combination of strategies that no player is willing to deviate from , and, for any player who does not choose the NE strategy, he or she will be punished for his or her deviation (as in "Games of Strategy, Second Edition (Dixit & Skeath, 2004)."), the ability to track down NEs in a complex system in advance shows certain intelligence in a autonomous system. By this logic, since Deep Deducing, in the preliminary result, tracks down NE in the prisoner's dilemma, there is certain intelligence in Deep Deducing. The pseudo code here

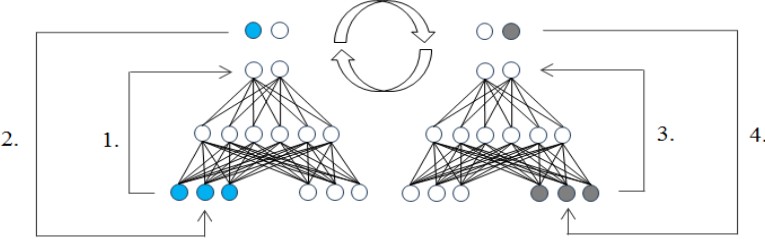

0. Initially randomize neurons that represent strategy input for player A and B
1. Generate output from current strategy input for player A and B obtained in step 0 or step 4 in the last epoch
2. Propagation the error between [1, *] and the generated output back to strategy input for player A. Keep strategy input for player A and B
3. Generate output from current strategy input for player B and A obtained in step 2
4. Propagation the error between [*, 1] and the generated output back to strategy input for player B. Keep strategy input for player A and B and then repeat from step 1

*** The desired payoff for player A and B and the weight of the synapse are all intact and left unchanged during all the epochs ***

| times | A strategy | B strategy | actual payoff for a round | desired payoff |
|---|---|---|---|---|
| 0 | [0.4, 0.5, 0.5] | [0.4, 0.5, 0.5] | [0.7, 0.5] | [1, *] and [*, 1] |
| 100 | [0.5, 0.5, 0.5] | [0.5, 0.5, 0.5] | [0.6, 0.5] | [1, *] and [*, 1] |
| 200 | [0.6, 0.5, 0.3] | [0.6, 0.5, 0.3] | [0.6, 0.5] | [1, *] and [*, 1] |
| 300 | [0.8, 0.6, 0.1] | [0.8, 0.6, 0.2] | [0.6, 0.5] | [1, *] and [*, 1] |
| … | … | … | … | … |
| N | [0.9, 0.7, 0.1] | [0.9, 0.7, 0.1] | [0.5, 0.5] | [1, *] and [*, 1] |

Figure 5: example of the deduction process for prisoners

can also be dissected into two phases, namely the training or learning phase and the deduction phase. For the training phase, we simply duplicate the training phase in the mosquito example except that this time the environment input was replace by strategy input for player B, and the original strategy by strategy input for player A.

# 4 EXPERIMENT RESULTS

To thoroughly testify the functionality of Deep Deducing, we will first permute the payoffs in the prisoner dilemma in Table 1 to check if Deep Deducing with a set of trained synapses can track down Nash Equilibriums (NEs) in different locations with arbitrarily randomized starting points for strategy inputs for player A and B in each permuted table. Then we will generate 50 game tables with the size of 2x2 and 4x4, with randomly generated payoffs, to check if Deep Deducing with a set of trained synapses can track down the sole and the only NE in each table with different size. The program we set up guarantees there is only one NE in each table. Therefore, at a random guess, the chance to spot NE is 25 % in table 2x2 and 6.25 % in table 4x4.

Preliminary result shows that Deep Deducing can track down NE in different locations with arbitrarily randomized starting points for strategy inputs for player A and B. Also the result shows that Deep Deducing can track down NEs at the precision of nearly 100 % in tables 2x2, and 70 % in tables 4x4, which is obviously higher than random guess. The whole source code corresponding to each Figure can be seen at: `https://github.com/AnonymousDeepDeducingCode/DeepDeducing/tree/master`, open for inspection.

The model here we will use is simply the traditional canonical 3-layered deep feedforward neural network with no modification. The squared error is used for error function (loss or cost function). To avoid noise in Back Propagation in Deep Deducing, the strategies are represented in digits. For example, strategies in table 2x2 can be represented by 1 digit, [0] or [1], since 1 digit is sufficient to represent all possible strategies for a player. Strategies in table 4x4 can be represented by 2 digits, [0, 0], [1, 0], [0, 1] and [1, 1], for each player. Also, the payoff for each player in each table is represented by digits as well. For example, payoff 3 is represented by [1, 1, 1, 0, 0, 0, 0, 0, 0, 0] with possible maximum payoff 10. The sequence is irrelevant and can be changed at user's will because the volume of error sent back to the input layer is all that matters. Here, we set maximum payoff as well as the optimal desired payoffs to 10 for each player. The final strategy adopted by a player depends on the closest range of its final strategy input to [0] or [1]. For example, if the strategy input for player A arrives at [0.13] in the end, player A will adopt strategy [0]. If the strategy input for player B arrives at [0.653] in the end, player B will adopt strategy [1].

The size of the 3-layered neural network is set to be n*h*m, where n is the sum of the size of the strategy inputs for A and B in each table, h is the size of hidden layer, and m is the sum of the size of the neurons that represent possible maximum payoffs for player A and B in each table. Therefore, n*h*m = (1+1)*h*(10+10) in table 2x2, and n*h*m = (2+2)*h*(10+10) in table 4x4.

For the first test, the prisoner's dilemma in Table 1 is permuted into 4 different tables as below in Table 2.

| | | B Strategies | | | | | B Strategies | |
|---|---|---|---|---|---|---|---|---|
| | | [0] | [1] | | | | [0] | [1] |
| A Strategies | [0] | [5, 5] | [10, 0] | | A Strategies | [0] | [7, 7] | [0, 10] |
| | [1] | [0, 10] | [7, 7] | | | [1] | [10, 0] | [5, 5] |
| | | B Strategies | | | | | B Strategies | |
| | | [0] | [1] | | | | [0] | [1] |
| A Strategies | [0] | [10, 0] | [7, 7] | | A Strategies | [0] | [0, 10] | [5, 5] |
| | [1] | [5, 5] | [0, 10] | | | [1] | [7, 7] | [10, 0] |

Table 2: permuted tables of prisoner's dilemma (with NE underline)

With a 3-layered deep feedforward neural network with size (1+1)*30*(10+10) with learning and deducing rate at 0.01 and learning and deducing epochs at 10000, the trajectories of arbitrarily randomized starting points for strategy inputs for player A and B in each corresponding table in Table 2 can be shown in the left part of Figure 15.

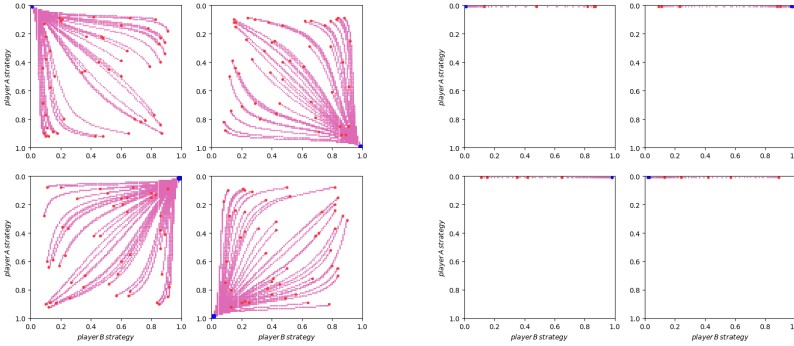

Figure 6: trajectories of strategy inputs for player A and B corresponding to Table 2 (left) trajectories of strategy inputs for player B under fixed strategy corresponding to Table 2 (right)

As we can see in the left part of Figure 15, regardless of the initially randomized starting points (denoted red dots) for the strategy inputs, the strategy inputs for player A and B converge to NE (denoted blue dots) in each table under Deep Deducing.

If We further fix the strategy input for player A to [0], and tune only the strategy input for player B, we will obtain the right part of Figure 15. As we can see in the right part of Figure 15, the strategy inputs of player B still converge to the optimal (and conditional) strategies for player B in Table 2 (denoted blue dots), regardless of its initially randomized starting positions (denoted red dots).

***Since a canonical deep neural network use sigmoid as activation function, when the training of the neural network is sufficient, a little deviation in the input layer does not generally affect the output of the neural network as a whole, which renders the optimization geometry for the strategy inputs rather smooth and further lessens the chance of getting stuck at local minimum for gradient descent and other types of optimization skills***. If we take sum of the squared errors $(\mathbf{o}^{dB} - \mathbf{o}'^{B})^{2}$ as axis z, we can obtain Figure 17.

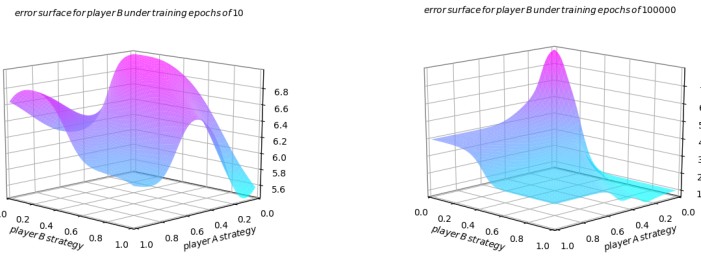

Figure 7: the learning or training process to smooth the optimization geometry for player B under lower left table in Table 2

We can see the smoothness of geometry for the strategy inputs increases as the neural network learns more about a game table and further lessens the chance of getting stuck at local minimum for gradient descent and other types of optimization skills (see Figure 9). This explains why Deep Deducing can be attached to Deep Learning.

However, how did the strategy inputs for player A and B automatically track down NE in the left part of Figure 15? This paper proposes two possible explanations. The first possible and more vivid explanation is that, imagine strategy inputs for A and B are two small ants in two different mountain sides, they both want to climb to the lowest points of their own mountains, however they overlap each other (phantom space) and wherever ant A moves, ant B will be carried to the same direction due to the this phantom effect, and vice versa. ***If ant A moves first and then ant B moves afterwards repeatedly, in the end, ant A and B will pull each other to a certain point (as the blue rectangles in Figure 8), and this point is never the lowest point of their own mountain side***. This can be shown in Figure 8.

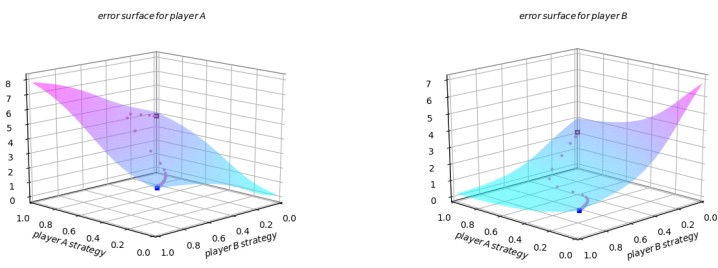

Figure 8: the trajectories of strategy inputs under different error surface for player A and B under lower left table in Table 2

The second possible explanation is that, In Kakutani's fixed point theorem, a topology space kept mapped into itself (if closed, continuous) will converge to a point. This point is the fixed point under every mapping. By Nash theorem, if the mapping is the pursuit of the best interest of each player, this point will be the Nash Equilibrium (as in "Introduction to Topology: Pure and Applied (Adams & Franzosa, 2008)."). In Figure 15, we can observe that the trajectory of strategy inputs for player A and B under Deep Deducing algorithm converge to a certain point regardless of their initial different position. Therefore, by Kakutani's fixed point theorem, there is a fixed point under every mapping. Further by Nash theorem, the fixed point shall also be the Nash Equilibrium since Deep Deducing mimics the rationale of human player in the pursuit of their own interest. Therefore, Deep Deducing tracks down the fixed point, and, therefore tracks down the Nash Equilibrium.

For the second test to track down NEs in 50 randomly generated game tables, we instead use a 3-layered deep feedforward neural network with size (1+1)*15*(10+10) for table 2x2 and (2+2)*15*(10+10) for table 4x4 with learning and deducing rate at 0.1 and deducing epochs at 5000. The success rate to track down NEs in each table type (2x2 and 4x4) varies with learning or training epochs. The final result is shown in Figure 9.

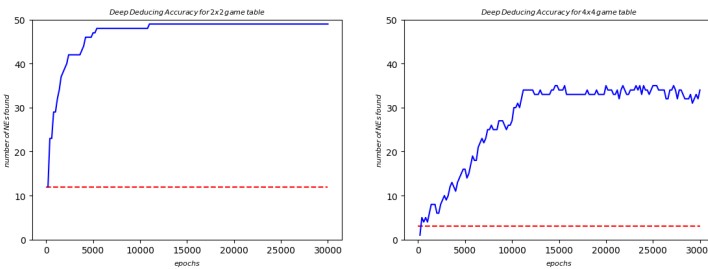

Figure 9: accuracy for tracking down NEs in different learning epochs

As we can see, Deep Deducing is almost 100 % accurate in tables of size 2x2, however the precision in tables of size 4x4 drops to 70%. ***This is because the learning process only lessens the chance of getting stuck at local minimum, we are only using plain and modest gradient descent with Back Propagation, and the strategy inputs for player A and B might get stuck at "common" local minimums in 4x4 tables, which minimum is the common minimum in the error surface of player A and B respectively (so we call it "common") in which case we cannot expect ant A and B to pull each other out of their local minimums***. To solve this problem, we will apply a simplified gradient-based particle swarm optimization and transform Equation 2 into Equation 3a and 3b for player A and 4a and 4a for player B:

$$\mathbf{m}_t^A \leftarrow \mathbf{i}_t^A - \mathbf{i}_{t-1}^A \tag{3a}$$

$$\mathbf{i}_{t+1}^A \leftarrow \mathbf{i}_t^A - \alpha \frac{\partial}{\partial \mathbf{i}_t^A} E(\mathbf{o}^{dA} - \mathbf{o}_t^{'A}) + \alpha \mathbf{m}_t^A + \alpha(\mathbf{i}_t^{A_{optimal}} - \mathbf{i}_t^A) \tag{3b}$$

$$\mathbf{m}_t^B \leftarrow \mathbf{i}_t^B - \mathbf{i}_{t-1}^B \tag{4a}$$

$$\mathbf{i}_{t+1}^B \leftarrow \mathbf{i}_t^B - \alpha \frac{\partial}{\partial \mathbf{i}_t^B} E(\mathbf{o}^{dB} - \mathbf{o}_t^{'B}) + \alpha \mathbf{m}_t^B + \alpha(\mathbf{i}_t^{B_{optimal}} - \mathbf{i}_t^B) \tag{4b}$$

where $\mathbf{i}_t^{A_{optimal}}$ is the optimal solution in the strategy swarm for player A at time t, and vice versa. Since Deep Deducing exploits Back Propagation as the gradient descent method, $\frac{\partial}{\partial \mathbf{i}_t^B} E(\mathbf{o}^{dB} - \mathbf{o}_t^{'B})$ can serve as a gradient for swarm optimization.

We initiate only seven particles in each strategy swarm A and B separately. The deducing epoch is set to be 100*100, where 100 is the iteration inside each swarm and 100 between swarm A and swarm B.

In the beginning of iteration inside a swarm, we keep the leader obtained from the last iteration and renew the rest of swarm (randomly) as well as all the momentum. In the mid of the iteration inside the swarm, the particles will learn from each other and improve their positions. And in end of the swarm iteration, the best strategy input will survive and be elected as the new leader and the rest of the swarm will be killed. This leader of swarm A will become a variable to consider with, for the iteration inside swarm B. The same process go through the strategy swarm of A and B. ***We can simply imagine this process as the democratic election system and international competition system***. By doing so, we exploit the benefit of gradient-based particle swarm optimization to escape local minimums while keeping the swarm as resilient as possible by introducing new particles to compete with (and learn from) the old leader, killing less efficient particles in the swarm, and then raising a possible new leader to confront the enemy swarm. We will then obtain Figure 10 as the renewed accuracy for 4x4 tables:

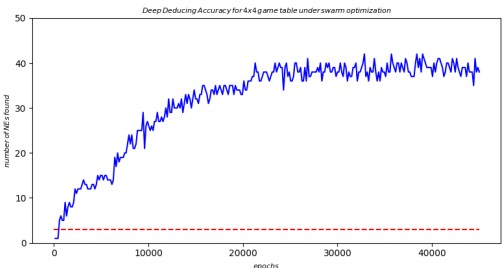

Figure 10: accuracy for tracking down NEs in tables 4x4 in different learning epochs coupled with simplified gradient-based swarm optimization

As we can see in Figure 10, the accuracy improves to 80 % for 4x4 tables . Deep Deducing tracks down NEs despite the fact that there is only one uncoupled algorithm.

For more interesting result, please consult Appendix. From evidence above (including Appendix), we can conclude that there is certain intelligence in Deep Deducing.

## 5 CONCLUSIONS

The experiments shown here are not all inclusive, but the preliminary experiments indicate that Deep Deducing can perform the similar task to human deduction by simply propagating error further back to the input layer. In this sense, the input layer of a neural network does not only serve as an information receiver (as we always envisage) but also an information sender, which information is further sent to other parts of the body such as muscles or even other neural networks.

The whole traditional deep neural network can be viewed as a gigantic memory or knowledge reservoir of strategy, environment and payoff. In a word, a simple traditional deep neural network itself is already a memory pool, ready to have its information be extracted for later usage, and Recurrent Neural Network just augmented this function by allowing variable inputs to be considered. This finding might implement the notion of "memory" as we envisaged (as in "Deep Learning (LeCun et al., 2015).").

By back-propagating error back to the input layer, a traditional deep neural network forms an internal information exchange route similar to the internal information exchange in Hopfield Neural Network. The only difference is that the former uses Back Propagation algorithm while the latter exploits the notion of thermodynamics (Hopfield Neural Network can be viewed as a Restricted Boltzmann Machine with visible layers mapping to itself in supervised learning, while a Boltzmann Machine can be viewed as a Hopfield Neural Network with hidden layers and simulated annealing, the latter see "Connectionist architectures for Artificial Intelligence (Fahlman & Hinton, 1987)."). This internal information from other pathways does not only train the synapses itself as we always envisage (as in "How neural networks learn from experience (Hinton, 1992)."), but also the source (neurons in the input layer) as well.

There are pros and cons to Deep Deducing. First, Deep Deducing is highly intuitive and can be easily coded. Second, Deep Deducing can be seamlessly combined with other types of neural networks as long as the back-propagation algorithm or back-propagation-like algorithm is compatible with the latter. However, the downside is that Deep Deducing requires heavy epochs as well as training time on the training or learning phase in order to achieve high precision in the deduction phase in some application.

There is still an abundant discovery that we have made, and we are still making progress.

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

## APPENDIX I (EXPERIMENT II): DEEP DEDUCING PLAYING TIC-TAC-TOE GAME IN A SEQUENTIAL DISCRETE GAME

The experiment so far touches upon only the combination of Deep Deducing and Deep Feedforward Neural Network in Simultaneous Discrete Game.

However, there are still Recurrent Neural Network in A.I. and Sequential Discrete Game in Game Theory, and we can still combine Deep Deducing and Recurrent Neural Network to play with Sequential Game.

Imagine a simple recurrent neural network with one fully connected input layer, hidden layer and output layer, and the interconnection between the hidden layers is still fully connected. Each layer is represented by a set of neurons marked in the shape of rectangle.

First, in the training or learning phase, we will train the RNN according to a game tree (shown in Figure 11) in Sequential Discrete Game in Game Theory. The whole training or learning phase is shown in Figure 12. This game tree may be generated by fixed program in a chess game or by trail-and-err in the real world like the case of mosquito.

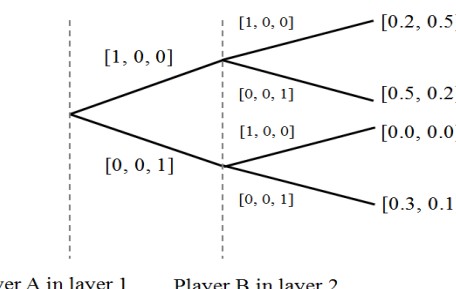

Figure 11: game tree for sequential game

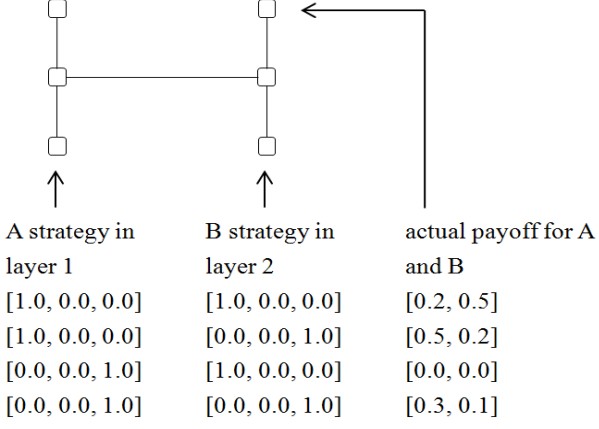

Figure 12: training or learning phase in Sequential Discrete Game

In the deduction phase, we first randomize strategy input for A and B. Then, by using the notion of "Back Induction" in Game Theory (as in "Games of Strategy, Second Edition (Dixit & Skeath, 2004)."), we tune the neurons in the strategy input for player B in layer 2 to meet the desired output [*,1] for player B, and then we tune the strategy input for player A in layer 1 to meet the desired output [1,*] for player A. At last we repeat the process again, and the strategy inputs obtained in the last epoch will be kept to the next epoch. The whole procedure can be shown in Figure 13.

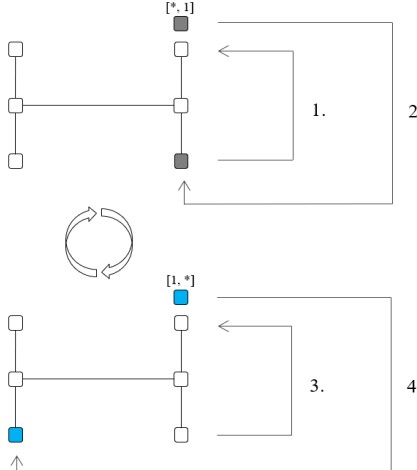

0. Initially randomize neurons that represent strategy input for player A and B
1. Generate output from current strategy input for player A and B obtained in step 0 or step 4 in the last epoch
2. Propagation the error between [*, 1] and the generated output back to strategy input for player B. Keep strategy input for player A and B
3. Generate output from current strategy input for player A and B obtained in step 2
4. Propagation the error between [1, *] and the generated output back to strategy input for player A. Keep strategy input for player A and B and then repeat from step 1

*** The desired payoff for player A and B and the weight of the synapse are all intact and left unchanged during all the epochs ***

Figure 13: deduction phase in Sequential Discrete Game

Finally, the final strategy input for player A in layer 1 in figure Figure 13 is the optimal strategy for player A in layer 1 in the real world. After player A chooses its strategy, in the next layer of the game in the real world, player B can exploit the same technic to obtain the optimal strategy for himself or herself in layer 2. Both players will converge to the Nash Equilibrium (route) in Sequential Discrete Game in Game theory in the end. The same methodology also applies to Sequential Discrete Game with arbitrary numbers of layers.

By this method, we can teach a RNN to learn to play Tic-Tac-Toe game. The experiment result shows that Deep Deducing can handle Tic-Tac-Toe Game. The whole source code can be seen at: `https://github.com/AnonymousDeepDeducingCode/DeepDeducing/tree/master`, open for inspection.

The model here we will use is simply the traditional canonical 3-layered (with one hidden layer fully interconnected) recurrent neural network with no modification. The size of the neural network is (9+9)*40*2. The squared error is used for error function (loss or cost function).

The program we set up will record historic strategies of the players and automatically generate the future legal strategies of the players to the end of the game as well as their final payoffs. The strategy of a player is represented by 9 digit corresponding to its position in the game board when flattened. The payoff for a player is simple, [1] or [0], meaning winning or not winning (might be draw).

When the recurrent neural network is trained according to Figure 12, the recurrent neural network will take history into consideration and deep-deduce the current optimal strategy according to Figure 13 in a pre-fixed future (here we set it to be the steps toward the end of the game). The final current strategy is the argmax of the deduced current "legal" strategy corresponding to its position in the game board when flattened.

The order of the game is that player A will move first.

We will let Deep Deducing RNN play with the human player and let the readers judge its intelligence by its states of the play. The states are recorded as the rows in Figure 14 (where "O" represents player A assumed by human and "X" represents player B assumed by Deep Deducing):

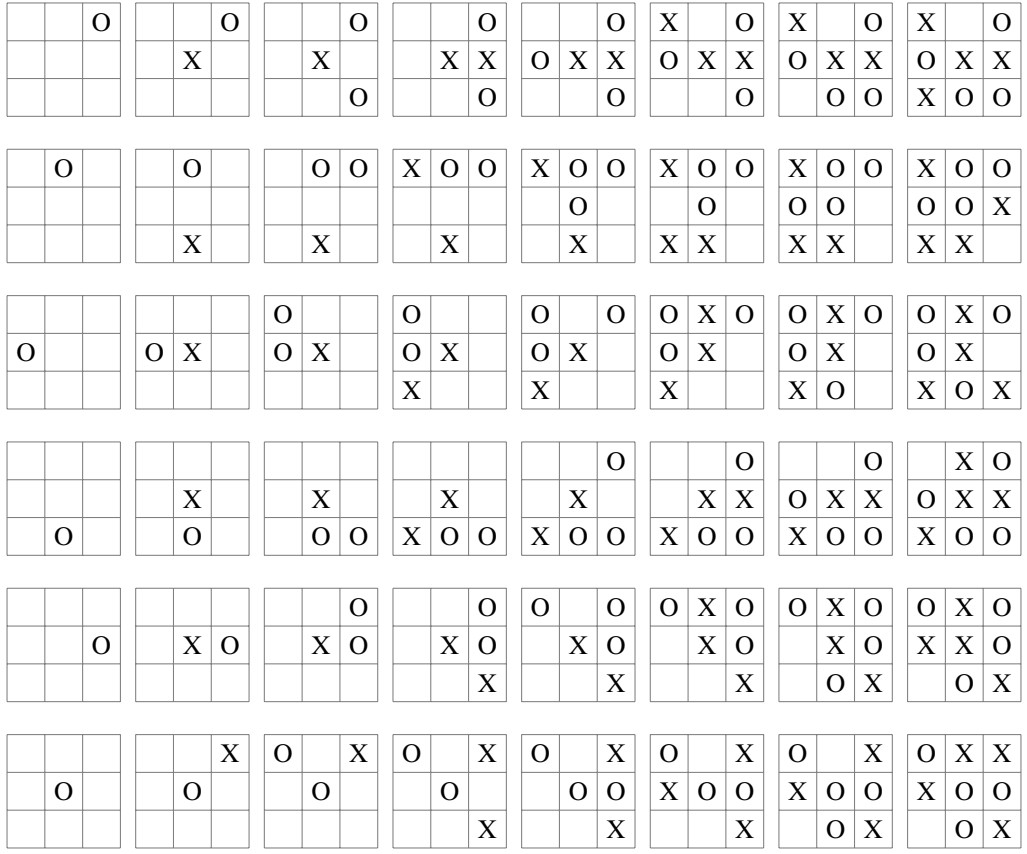

Figure 14: the states of the play in experiment with Deep Deducing

As we can see, if adequately tuned, in some tricky first-hand advantages, Deep Deducing can avoid bad moves in order to survive.

However, the curious thing is that if we set the number of steps foreseen in the future to be odds, Deep Deducing will take a aggressive stance. In a sense, it will not defend, and, In some cases, it leads to its defeat. If we set the number of steps foreseen in the future to be evens, Deep Deducing will take a defensive stance. In some cases, it misses its chance to win, although it won't be defeated. Therefore further improvement is required.

Nonetheless, the goal of the preliminary experiment is to indicate that Deep Deducing can perform the similar task to human deduction by simply propagating error further back to the input layer.

## APPENDIX II (EXPERIMENT III):: DEEP DEDUCING PERFORMING CROW REASONING, TRANSFORMING FEED-FORWARD NEURAL NETWORK INTO RECURRENT NEURAL NETWORK

Imagine a crow in the video: `https://youtu.be/cbSu2PXOTOc` was trained to solve different puzzles such as dropping stones into bottle to get food and picking up a key to unlock the cage to get food. The crow is familiar with each different puzzle and it was trained to solve each of the puzzles separately and non-sequentially. However, when faced with a consecutive puzzle that must be solved by dropping stones into bottle to get a key and then use the key to unlock the cage to get food, can the crow can still solve the puzzle? The answer is yes. The crow, though not knowing how to speak human language, can perform the reasoning task of an intelligent specie.

In this experiment, we will try to let Deep Deducing mimic the rationale of the crow. Since Deep Deducing exploits Back Propagation and propagates error back to the input layer, and the input layer of a neural network can serve as the output layer of another, for a single trained deep feed-forward neural network, we can stack the deep feed-forward neural network into a recurrent neural network and make a crow-like reasoning to solve a consecutive puzzle. This single trained deep feed-forward neural network resembles the brain of the crow after it being trained to solve each puzzle separately and non-sequentially. The DFNN-transformed recurrent neural network resembles the brain of the crow when faced with a consecutive puzzle, and the recurrent neural network must figure out a sequence of strategies to meet the goal, such as dropping stones into water to get the key and then using the key to get the food. The whole process can be shown in Figure 15:

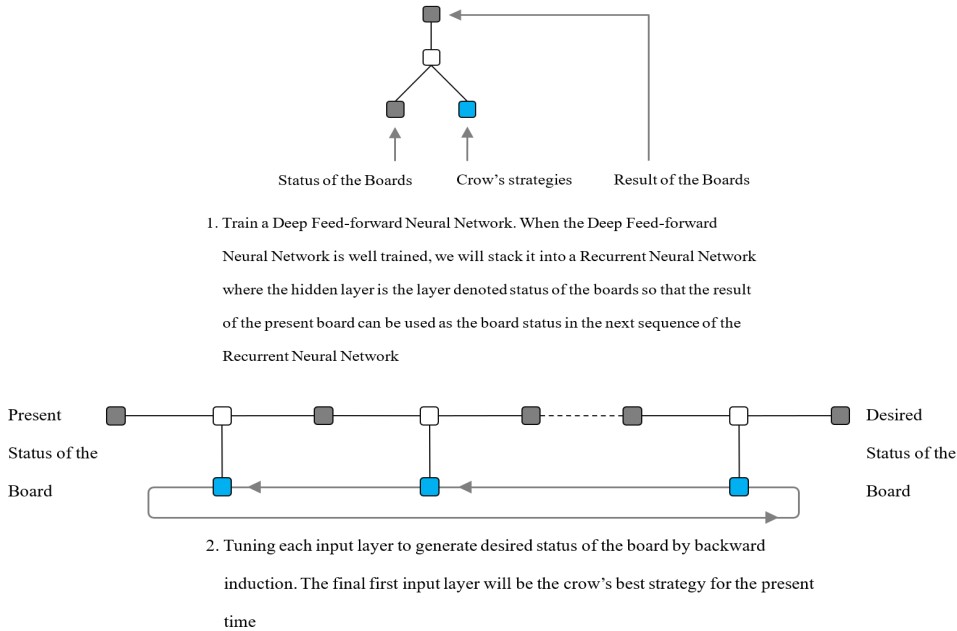

Figure 15: transforming Feed-forward Neural Network into Recurrent Neural Network to perform crow reasoning by Deep Deducing

To testify the functionality of this method, we will start from teaching a machine to move points A to points B in a 5x5 grid. We first train a deep feed-forward neural network to learn what position in the grid the points will achieve when it pushes the present points to left, up, right, down or remains the points still. Then we will permute the deep feed-forward neural network into a recurrent neural network by Figure 15, making the hidden as well as the output layers as the states of the grid and the input layer as the direction where it moves the points. Then we set the position of points B as the goal where the machine is expected to push points A to match the status of points B. By deep deducing, the machine starts a crow-like reasoning and find the shortest route to push points

A to where points B are located. The whole source code can be seen at: `https://github.com/AnonymousDeepDeducingCode/DeepDeducing/tree/master`, open for inspection. Part of the results is shown in Figure 16 (to avoid confusion with points A, the position of points B is shown only once in the first grid, and if A overlaps B, only B will be shown):

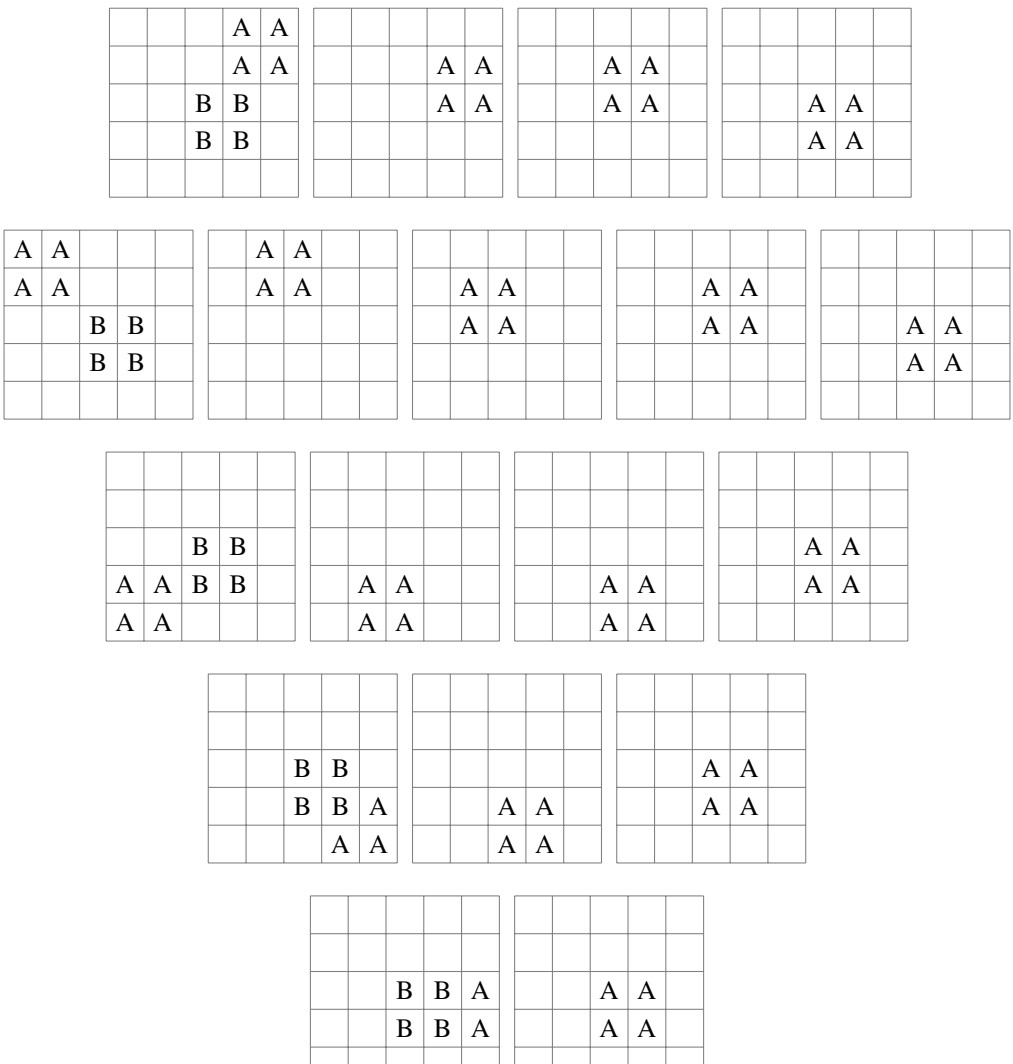

Figure 16: the recorded states of the movement of DFNN-transformed RNN under Deep Deducing

Interestingly, for a pre-fixed number of steps taken into consideration in the future (which need not to be the exact shortest number of steps to where points B is located), the machine mimics the rationale of a crow and moves points A to the location of points B in the shortest route. The machine can choose to remain still to avoid wasting unnecessary steps (to save space, only "still" strategy is not recorded in Figure 16). The DFNN-transformed RNN, when back-propagating error back to the input layers sequentially, can make sequential deduction and combine the knowledge pieces learned from the environment into a logical, functional and sequential reasoning to achieve its goal, which is way too similar to that of a crow.

However, smart readers might have noticed that, the crow's consecutive puzzle is not to drop stones into bottle to get "food" and then use the "food" to unlock the cage to get food, but to drop stones into bottle to get "key" and then use the "key" to unlock the cage to get food, which means the initial environment that the crow faces is different from its training environment.

So the further question is that, can a DFNN-transformed RNN, under Deep Deducing, perform the same reasoning under initial environment with partial noise (such as the food in the bottle was replaced with a key)? Let us try it out. We first add some noise in the present status of the board and see if a DFNN-transformed RNN under Deep Deducing can perform the same reasoning. The result is shown in Figure 17:

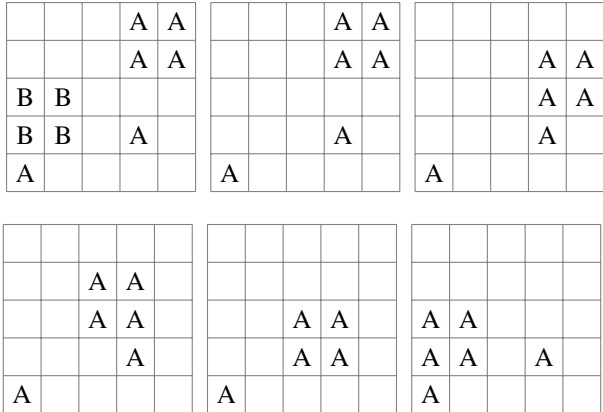

Figure 17: the recorded states of the movement of DFNN-transformed RNN under Deep Deducing with partial noise

Interestingly, DFNN-transformed RNN under Deep Deducing with partial noise can still find a route to push points A to points B despite that fact that there are noise in the initial status of the board. And more interestingly, sometimes this route might not be the shortest route (as shown in the second status in the first row in Figure 17 where the machine took an unnecessary move to move the points A to right direction to hit the wall), which means that, if there are noise in the initial status, Deep Deducing might take a longer detour to reach the goal, which is way too similar to the crow in the video above, where the crow might not succeed in the first try.

Nonetheless, the goal of the preliminary experiment is once again to indicate that Deep Deducing can perform the similar task of intelligence deduction by simply propagating error further back to the input layer.

