# OpenReview forum: "FROM DEEP LEARNING TO DEEP DEDUCING: AUTOMATICALLY TRACKING DOWN NASH EQUILIBRIUM THROUGH AUTONOMOUS NEURAL AGENT, A POSSIBLE MISSING STEP TOWARD GENERAL A.I."
_ICLR.cc/2019/Conference_

### Official Review · AnonReviewer3 · 2018-11-05
**Not good enough**

**Rating:** 4
**Confidence:** 5

**Review:**

This paper tries to build a neural net to learn Nash equilibrium of games, even though it has been proved that no uncoupled algorithm can do that, except on specific games, as the ones considered in the example (0-sum, potentiel, solvable by iterated elimination of dominated strategies, etc.).

The algorithm proposed is a classical neural net without any insight (I believe its behavio must more or less be similar to regret matching).

In table 10, I do not think that the underline case is the NE.

In table 12, the algorithm si conveniently initiated at the NE.

---

### Official Review · AnonReviewer2 · 2018-11-08
**weak experimental evidence and unconvincing arguments**

**Rating:** 2
**Confidence:** 4

**Review:**

The paper presents an approach to infer optimal strategies by learning the payoff function of 2 player games with a neural network Q(s, a), where a are the agent actions and s the context (e.g., action of the other player). The strategy is inferred by considering the inputs as free variables at test time and maximizing the learnt Q over its input variables by backpropagation.

The structure of the neural network in itself is not particularly original. The originality of the paper is to show that experimentally, in some 2-player games (and small sequential games, using an RNN), the policy that is inferred is close to a Nash equilibrium. While this is an interesting result in itself, the games used in the experiment are pretty easy to solve with existing algorithms, so the experimental evidence that this approach can work in practice in difficult cases is weak.

The motivation and intuition fail to be convincing. There is an excessive usage of analogies with intelligence and life in general that are not particularly enlighting (e.g., "Even though its hardware is damaged, the software in
the cloud (or the eternal soul, arguably) of the mosquito [...]", the value of the analogy between the cloud and the soul is unclear), and in the end there is no clear explanation of why it should work in practice.

I think the paper in its current form is not ready for publication. More formal arguments and/or stronger experimental evidence are necessary.

---

> ### Author Response · Authors · 2018-11-12
> **Thank you, reviewer 2**
>
> Thank you for your opinion. Indeed, there is much need to be done. We agree with your opinion.
>
> (1) The name of this machine is inappropriate since we didn't really invent a new kind of neural network, and we only exploit back propagation in an nontraditional way. We will cherish your opinion and re-coin the name of the machine.
>
> (2) The experiment result is not enough, indeed. We are working on the experiment result. The experiment result requires some time to come by (if there are randomly generated 100 tables, it takes quite a time), and we are working on it.
>
> Thank you again for your time reviewing this paper. We sincerely thank you for your time and precious opinion. Thank you again.

---

### Official Review · AnonReviewer4 · 2018-11-26
**Poorly written paper with preliminary experiments**

**Rating:** 3
**Confidence:** 3

**Review:**

The paper proposes a method of searching for a Nash equilibrium strategy in games where the strategy-to-payoff mapping is defined by a neural network. The idea is to perform gradient optimization of the payoff w.r.t. the strategy. Preliminary results on tic-tac-toe and variations of the prisoner’s dilemma task are presented. The paper has an interesting idea at the core. However, it is poorly written, does not properly discuss the related works and does not present a convincing method or experimental results.

Pros:
* The paper considers an interesting question of exploring the applications of neural networks to the game theory problems.
* The idea of the paper is reasonable. It makes sense to me to perform gradient-based search over the strategies (assuming that the payoff is differentiable).

Cons:
* Writing
- The paper is over the mandatory length limit of 10 pages.
- The paper makes a grandiose claim: “this paper provides a revolutionary way for reinforcement learning and a possible road toward general A.I.” However, there are arguably no revolutionary ideas, and certainly no reinforcement learning experiments!
- Despite the claim, the novelty of the paper is limited. There is no discussion of the related work: optimization of the neural networks w.r.t. the inputs [1]; related RL ideas such as model-based learning [2,3] and Monte-Carlo Tree Search [4].
- The problem being solved is never formally stated. As far as I understand, Nash equilibrium is usually defined (1) in mixed strategies, while the paper seems to consider pure strategies; (2) in the scenario where every player attempts to maximize their payoff, while in the paper the players attempt to achieve some pre-fixed value of the payoff.
- The flow of the paper is generally poor. Instead of presenting a general solution and then showcasing its applications, the paper iterates on similar ideas multiple times. For example, all four algorithms are just gradient-based optimization of either the weights or the inputs to a model.
- The paper provides extremely misleading analogies and explanations. I am quite sure that a mosquito brain is not a one hidden layer fully-connected neural network! Also, the example of avoiding a moving hand is poor: since the outcome is life or death, the learning should happen via evolution, not during the lifetime of a single insect. The claim that the neural networks with sigmoid activation functions are less prone to local optima is questionable as well.

* Method and experiments
- The proposed method is essentially a greedy gradient-based planning procedure. For this to work, we need to have a very good environment model. This is a strong assumption that is not discussed.
- The experiments are performed on very simple synthetic problems: matrix games and tic-tac-toe. They do not suggest that the method is general and can work on harder problems, say, Sokoban [2].
- The experiments do not present any baselines, so it is unclear how well the method performs compared to the alternatives. One obvious candidate is gradient-free optimization, such as Nelder-Mead, and gradient descent with momentum, which can be less prone to local optima.

[1] Brandon Amos, Lei Xu, J. Zico Kolter “Input Convex Neural Networks”, ICML 2017
[2] Racanière et al. “Imagination-Augmented Agents for Deep Reinforcement Learning”, NIPS 2017
[3] David Ha, Jürgen Schmidhuber “Recurrent World Models Facilitate Policy Evolution”, NIPS 2018
[4] Thomas Anthony, Zheng Tian, David Barber “Thinking Fast and Slow with Deep Learning and Tree Search”, NIPS 2017

---

### Meta-Review · Area_Chair1 · 2018-12-13
**Not yet ready for publication**

**Confidence:** 4
**Recommendation:** Reject

**Metareview:**

The paper presents "deep deducing", which means learning the state-action value function of 2 player games from a payoff table, and using the value function by maximizing over the (actionable) inputs at test time.

The paper lacks clarity overall. The method does not contain any new model nor algorithm. The experiments are too weak (easy environments, few/no comparisons) to support the claims.

The paper is not ready for publication at this time.